# Magnetic Resonance Imaging Virtual Histopathology from Weakly Paired Data

**Anonymous**

**Editor:** NA

## Abstract

The pathological analysis of biopsy specimens is essential to cancer diagnosis, treatment selection and prognosis. However, biopsies are only taken from part of the tumor and cannot assess the full cellular extension. Such information is essential to delineate as accurately as possible the tumor volume on a three-dimensional basis. Furthermore, they require highly qualified personnel and are associated with significant risks. The aim of our work is to provide alternative means to gather clinical information related to histology through MR image translation towards virtual pathological content generation. Conventional approaches to address this objective exploit paired data that is cumbersome to achieve due to tissue collapse and deformation, different resolution scales and absence of plane correspondences. In this paper, we introduce a versatile, scalable and robust closed-loop dual synthesis concept that composes two generation mechanisms - cycle-consistent generative adversarial networks -, one exploring weakly paired data and a subsequent harnessing virtually generated paired correspondences. The clinical relevance and interest of our framework are demonstrated in prostate cancer patients. Qualitative clinical assessment and quantitative reconstruction measurements demonstrate the potential of our approach.

**Keywords:** Synthetic Histopathology, Generative Model, Whole Slide Image, Magnetic Resonance Imaging, Unsupervised Learning.

## 1. Introduction

The comprehensive understanding of tumor cellular environment is decisive yet scarcely accessible during cancer care. Ideally, such assessment comes from microscopic/genetic analysis of surgical resection, de facto absent prior to treatment. Alternatively, biopsies give substantial biological information but are invasive, costly, and the small harvested sample - when accessible - cannot catch tumor local heterogeneity. In addition to clinical data, oncologists thus rely on anatomical imaging such as computed tomography (CT) or Magnetic Resonance (MR) to perform diagnosis, prognosis, treatment and patient's follow up. Nevertheless, both modalities suffer from the lack of biological ground truth, resulting in highly variable decisions (Jager et al., 2016). The objective of our work is to lay a first stone towards easily available, automatic, non-invasive histopathological assessment. To this end, we propose virtual histology generation from MR imaging. The purpose is to help practitioners with complementary tool that infers cellular characterization of tumour infiltration, hence bridging the gap between biological and anatomical imaging.

Digital pathology has gained growing interest consequently to the generalized availability of scanned Whole Slide Images (WSIs). Applying computational approaches to this type of data can reveal rich phenotypic information. In this vein, deep learning methods have

been developed for tasks such as tissue classification, tumour segmentation or biomarker extraction towards survival models (Madabhushi and Lee, 2016; Bera et al., 2019). More recently, the advent of Generative Adversarial Networks (GANs) enhanced the potential of computational pathology, and more broadly medical image computing (Goodfellow et al., 2014; Yi et al., 2019). Quiros et al. (2020) developed PathologyGAN to generate realistic histological tiles and better understand morphological features of cancer lesions. GANs also suit domain adaptation, more particularly image-to-image translation, which is a transfer learning task where an image is transformed to fit the domain of one another. Pix2pix and cycleGANs frameworks for paired (and unpaired, respectively) images are predominant models to address it (Zhu et al., 2020; Isola et al., 2018). In digital pathology, studies mainly focused on stain transfer, where the objective is to use GANs to build stained-uniform datasets improving performance of downstream machine learning architectures (Srinidhi et al., 2019).

Quite surprisingly, few studies have tackled the radiology-pathology image translation aspect, while its potential for cancer treatment is universally recognized (Njeh, 2008). For instance, Shimomura et al. (2018) built a pipeline to infer pathological patch from MR voxel intensity but did not scale to WSI. Such sparse literature can be explained by the absence of paired images due to tissue collapse and the underlying extreme deformations between *in vivo* radiology and *ex vivo* specimen, leading to hardly usable data. In order to extract relevant correlations between both modalities, one first needs to establish plane correspondences between 2D WSIs and 3D volumes from radiology, which is eased by a 3D histological reconstruction (Ward et al., 2012). Next, deformable co-registration can be performed through either classical optimization algorithms (Chappelow et al., 2011) or deep learning models (Shao et al., 2020). Therefore, the creation of comprehensible, pixel-based, deep learning-compliant, paired images is burdensome and subject to successive uncertainties.

In this paper, we suggest to exploit generative models to predict histology from MR. In particular, the novelty lays on the use of weakly paired images on which unsupervised learning is performed. To do so, we build a synthesis pipeline made of two cycle-consistent GANs, the first one consisting in generating without supervision aligned ground truth histology, the second one improving synthesis on registered pairs in a self-supervised setting. For this preliminary study, we focus on the realism and accuracy of WSIs at MR-based resolution rather than trying to display them at real scale, which will be the primary focus of our future works. Therefore, we are fully aware that, because of resolution difference, precise biological content cannot be extracted from our generated WSIs yet, and the real purpose of our study is to take the first methodological step for MR-histology generation.

## 2. Methods

The study focuses on cancer prostate for which dataset is detailed in 3.1 and consists in weakly paired MR-histopathological 2D images. Our approach - depicted in Fig. 1 - relies on a recursive generation process that first adopts unsupervised generation through cycleGAN framework ($G$ in Fig. 1) to create synthetic histology training set (subsection 2.1), the latter being fed to the second self-supervised generative model (subsection 2.2). The output is a synthetic MR-registered histology, made available to the oncologist prior to treatment. Two

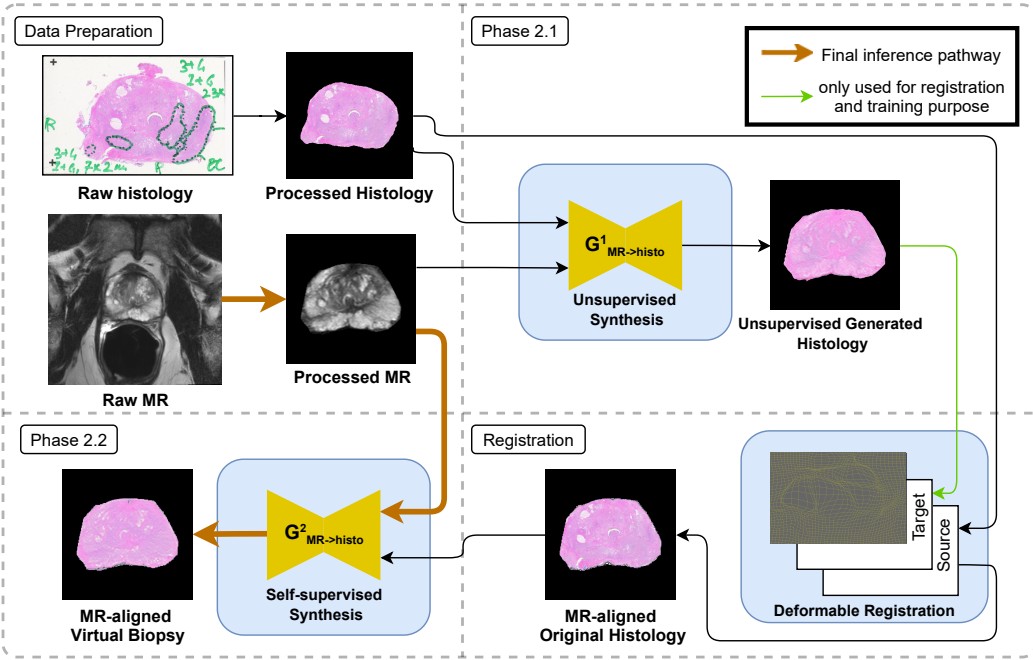

Figure 1: Overview of the two phase deep learning pipeline separated by a registration step and following preprocessing, with a sample of weakly paired images on prostate

additional sub-blocks, preprocessing and registration, make the whole pipeline autonomous. Once trained, only the inference pathway (orange arrow) is taken for straightforward virtual histology generation from MR.

## 2.1 Unsupervised generation on weakly paired data

The goal of the unsupervised generation is to build a paired dataset made of original MR and synthetic histology, only from weakly paired images as input. To do so, we use the cycleGAN framework, which consists in four deep neural networks trained in parallel (Zhu et al., 2020). Two of them are generators, the first one for histology generation from MR, the second one for MR generation from histology. The remaining two networks are called discriminators and try to distinguish between original and synthetic images. The detailed architecture is shown in supplementary material (Fig 5).

Discriminator is inspired from the PatchGAN architecture introduced by Isola et al. (2018) and focuses on real/fake classification for small patches of pixels to model texture-linked high-frequency structures and avoid blurry results. Generator is similar to the U-Net used in Pix2Pix, with the addition of residual blocks between the contracting and expanding paths. They act as skip connections to ensure stability in image reconstruction. The two generator architectures (respectively discriminators) are identical, the only difference being the input fed to them.

We note $G_H$ (resp. $G_{MR}$) the generator function from MR to histology (resp. from histology to MR). The corresponding discriminator functions are denoted $D_H$, $D_{MR}$. The

objective for the optimizer is a weighted sum of an adversarial loss between synthetic and original images for each modality, along with a L1 loss between cycle-reconstructed $(G_H(G_{MR}(h))$ for histology) and original images, and a final loss enforcing identity mapping and helping color generation when passing the generator an image from the target domain instead of the input domain. Each loss for histology is shown in equations (1,2), the same being applied for MR modality.

$$L_{adv}(h) = E_{h \sim p_{data}(h)}[\log(D_H(h))] + E_{mr \sim p_{data}(mr)}[\log(1 - D_H(G_H(mr)))] \qquad (1)$$

$$L_{rec}(h) = E_{h \sim p_{data}(h)}[||G_H(G_{MR}(h)) - h||_1], \ L_{Id}(h) = E_{h \sim p_{data}(h)}[||G_H(h)) - h||_1] \quad (2)$$

The total loss for both modalities is highlighted in equation (3) and the optimization goal is summed up by equation (4). The value of weights $\lambda_{rec}, \lambda_{Id}$ are chosen to be 10, to have balanced leverage between each component. For each loss, $L = L(h) + L(mr)$

$$L_{total} = L_{adv} + \lambda_{rec}L_{rec} + \lambda_{Id}L_{Id} \qquad (3)$$

$$G_H^*, G_{MR}^* = arg \min_{G_H, G_{MR}} \max_{D_H, D_{MR}} L_{total} \qquad (4)$$

The cycle architecture has been proven very helpful for unpaired generation compared to classical encoder-decoder or conditional GAN models. Indeed, reconstruction loss ensures cycle consistency, solving the pixel-wise loss issue for unpaired images. Nevertheless, we benchmarked simpler methods, ie encoder-decoder (only $G_H$) and conventional GAN (only $G_H$ and $D_H$) to quantitatively prove the benefit of cycleGAN.

### 2.2 Self-Supervised Training with Weakly Paired Correspondences

**Registration** The synthetic histopathological images from MR provide a more informative and discriminative space to perform deformable registration on original histological slices. Because the cycle generator $G_H$ outputs an image aligned on the original MR - it is easier for reconstruction to generate an histology with the same shape as the input-, we will thus obtain a post-process training set made of paired original MR/registered original histology. We can then see the previous step as an unsupervised training set generation. Aligning images from histological domain only is more effective than cross-modality registration, justifying the use of such deep learning step. The registration is run thanks to SimpleElastix library, with built-in Python methods (Lowekamp et al., 2015). It is made of an affine deformation as initialization, followed by a B-spline registration. We apply it on the whole training set to generate new paired dataset, which will become the input for the self-supervised learning method.

**Self-Supervised cycleGAN** The supervised pipeline is similar to the unsupervised one, the only difference being the pairing of input images. We have made the choice to keep the cycleGAN architecture despite the fact that it is no longer a theoretical necessity, but because results have been proven better even on paired data (Zhu et al., 2020). In order to help training, we added a L1 pixel-wise paired loss to the previous setting between the

generated and original images from same modality that are now aligned. Such loss for both modalities is shown in equation (5).

$$L_{paired} = E_{h \sim p_{data}(h)}[||G_H(mr)) - h||_1] + E_{mr \sim p_{data}(mr)}[||G_{mr}(h)) - mr||_1] \qquad (5)$$

The final loss is depicted in equation (6). We also chose 10 for $\lambda_{paired}$, and the optimization process remains the same as equation (4)

$$L_{total} = L_{adv} + \lambda_{rec}L_{rec} + \lambda_{Id}L_{Id} + \lambda_{paired}L_{paired} \qquad (6)$$

Because the data is paired, generators can learn faster with less parameters. We thus reduce the complexity of filters to make the model more scalable in terms of memory usage and speed.

## 3. Experimental Validation

### 3.1 Clinical Context and Data Preparation

The publicly available TCIA "Prostate MRI" dataset was used for the validation of our method (Choyke et al., 2016; Clark et al., 2013). It consists in 25 subjects who had a preoperative prostate MRI obtained with an endorectal and phased array surface coil. Each patient had biopsy and underwent prostatectomy. A mold was generated from each MRI, and specimen was first placed in the mold, then cut in the same plane as the MRI, into 3 to 6 slices.

We inspire from a study also using this cohort for complete registration (Shao et al., 2020). Plane correspondences were first established between 3D MR volume and 2D histopathology slices. Two additional operations were performed: (i) resizing all images to 420x420, that happened to be a fair balanced resolution between memory consumption and visual interpretability of results, and (ii) removal of manual pathological annotations of cancer presence that are deleterious for our generative task as it can be interpreted as organic tissue. We used color deconvolution - in particular RGB to Haematoxylin-Eosin-DAB (HED) - to extract and remove such annotation, and then recolored the pixels with interpolation. End-to-end, we obtain 83 processed pairs of images, being weakly paired yet unregistered.

### 3.2 Training/Inference details

Because of the small size of the dataset, we performed data augmentation prior to learning. More precisely, we combined rotations, in-plane translations, horizontal flipping and stain contrasting. We split the dataset into 50/10/23 pairs for training/validation/testing steps, being careful about bias issues when selecting both the distribution of patients and the level of slice into the volumes. Adam optimizer was selected for gradient descent calculus, with $\beta_1, \beta_2$ parameters equal to 0.5, 0.99 respectively. The learning rate was set to $2 \times 10^{-4}$, following guidelines from Zhu et al. (2020), with a linear decrease after half of training. We trained our model for 700 epochs for both unsupervised and self-supervised tasks, with a batch size of 2. Finally, we replace Binary Cross Entropy loss of discriminator with L2 loss to avoid saturation issues, as suggested by Mao et al. (2016). Various losses are displayed on supplementary material (Fig 6).

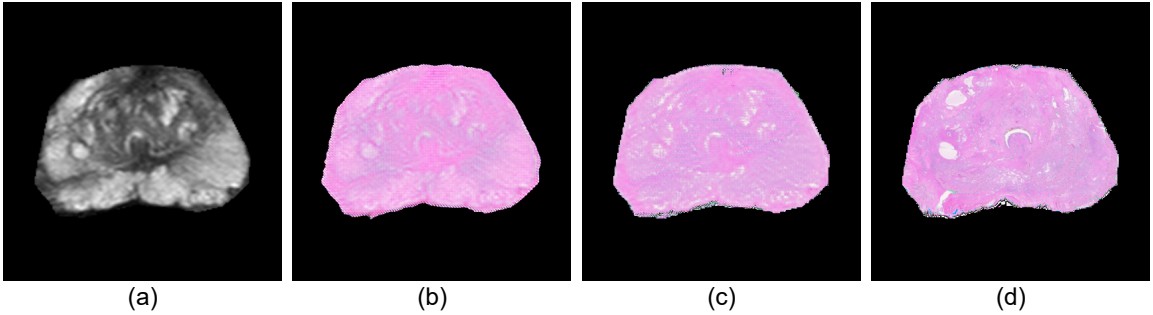

Figure 2: Sample from test set: ground-truth MR (a), unsupervised synthetic histology (b), final histology after self-supervised task (c), and ground-truth histology (d).

### 3.3  Results and Discussion

**Generalities** End to end, our model achieves a mean absolute error (MAE) of $5.9 \pm 1.7 \times 10^{-2}$ between the synthetic and the original registered histopathological slices on the held-out test set of 23 slices. A sample of such pairs is shown in Fig 2, along with the ground truth, and the unsupervised generated slice (phase 2.1) for visual comparison of improvement. Both generated images seem realistic, in terms of texture or shape, although smoother than original. Characteristic parts of prostate, such as urethra, central zone (CZ), transition zone (TZ), peripheral zone (PZ), and anterior fibromuscular stroma or anterior zone (AZ) are well generated on synthetic histology when visible on MR and ground truth histology. We study such anatomical landmarks with semi-quantitative assessment in Fig 3 and Table 2. One important precision, inherent to the data, is related to the empty-tissue blanks on histology. They are also present in ground truth but are seen by our model as a MR-intensity related particular type of tissue, whereas they are just the consequence of human manipulations during and after resection. This could lead to errors in biological interpretations and highlights the necessity of high-quality datasets for complete assessment of environment. Inversely, in many cases, inference from *in vivo* MR leads to a virtual histology without blank spots, thus reproducing a more reliable representation of biological environment than post-resection ground-truth histology. Finally, when focusing on the comparison between unsupervised and supervised generation, there is a substantial improvement in data quality, in particular in blurriness, confirmed by quantitative results. Other samples from test set are shown in supplementary material (Fig 4).

**Pixel-wise results** We computed quantitative metrics to justify our architecture and prove the robustness of the model, based on classical GAN performance measurement. The pixel-wise metrics are MAE and Peak Signal to Noise Ratio (PSNR) ; the structure-based metrics, assessing the realism of generated dataset compared to the original one, are Fréchet Inception Distance (FID) and Structure Similarity (SSIM). PSNR is an approximation to human perception of reconstruction quality, a higher score meaning closer datasets. In the same way, SSIM measures the structural degradation of a reconstructed image based on an original one but does not need registered images, and is scaled to $[0, 1]$, 1 meaning the datasets are the same. FID measures the distance between distributions of generated and original

Table 1: Quantitative results of our pipeline and comparison with simpler methods to prove the benefit of cycleGAN approach and two phase pipeline

| Metric | $G_H$ only | GAN only | cycleGAN only | Whole Pipeline |
|---|---|---|---|---|
| **MAE** ($\times 10^{-2}$) | $32.4 \pm 1.2$ | $10.2 \pm 0.9$ | $7.9 \pm 1.3$ | $5.9 \pm 1.7$ |
| **PSNR** | $3.06 \pm 0.38$ | $12.78 \pm 1.24$ | $12.93 \pm 1.18$ | $24.25 \pm 1.71$ |
| **SSIM** | $0.19 \pm 0.03$ | $0.67 \pm 0.02$ | $0.72 \pm 0.03$ | $0.84 \pm 0.03$ |
| **FID** | $358.5$ | $275.5$ | $188.7$ | $116.1$ |

sets as latent vectors, 1 meaning the datasets are the same. Formulas for these three metrics are explained in supplementary material.

The first question tackled the use of a cycleGAN on weakly paired data. It has been proven that such architecture is necessary to obtain decent results for image-to-image translation on unpaired setting because of the lack of ground-truth, and by consequence of piwelwise loss to enforce not just realism but also accurate correspondence. Nevertheless, to quantitatively assess this necessity, it is interesting to benchmark the method with simpler architectures, that is only the encoder-decoder $G_H$, and the conditional GAN $G_H$ with $D_H$. We compare them with the cycleGAN architecture on unsupervised setting. Based on the first three columns of Table 1, we can see a clear improvement in each metric, in particular with $G_H$ alone giving poor results because data is unpaired and the loss is pixel-wise.

The second question was related to the importance of both the second cycle and the registration process (phase 2.1 in Fig 1). Indeed, a more straightforward method would consist in only unpaired synthesis, giving MR-registered generated histology. Hence, we compared our results between the simple cycleGAN and the two successive ones. Based on last two columns of Table 1, the whole two phase pipeline outputs more realistic results than a simple cycleGAN on unsupervised setting, validating the intuition about easier learning on paired data mentioned above. For instance, the gain in MAE is 33.9%. Because the input is paired in the second setting, the generator can focus on texture and give more understandable images for the oncologist. In addition, FID, SSIM and PSNR scores are substantially enhanced, which is complementary to the qualitative visual results.

**Semi-quantitative results on anatomical landmarks** To qualitatively assess our work, we asked an expert pathologist to compare real and synthetic WSIs through a semi quantitative study on characteristic areas of prostate. Images could be categorized into four classes: (i) zones are well identifiable, (ii) moderately, (iii) hardly and (iv) unidentifiable. Basically, the distinction was made upon the number of zones the pathologist could delineate, (i) being all and (iv) being none of them. On real WSIs, difficulties of segmentation either come from the presence of a tumor deforming tissues, or from the level of the slice being out of the scope for a particular zone. Nevertheless, the real interest of such a study is to assess the degradation of generation made by the model on biological tissue, and not only the proportion of well generated pseudo WSIs - see first two rows of Table 2. Hence, we built three new categories: zones on pseudo-images are as well identifiable as on real images; the degradation corresponds to the shift from one class to at most the next one, or the degradation is important (gap $\geq 2$ between classes). Overall, images are not degraded

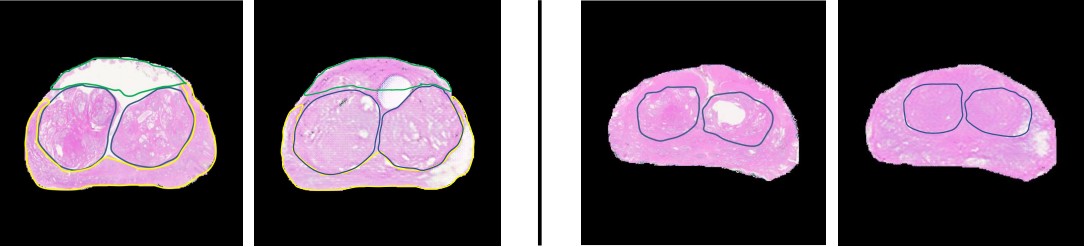

Figure 3: Delineation of prostate areas from pathologist on both real (left) and synthetic (right) images. For these two examples, AZ is in green, PZ in blue, TZ in yellow

Table 2: Semi-quantitative results on generation quality and comparison with real WSIs

| Image | Well | Moderately | Hardly | Unidentifiable |
|---|---|---|---|---|
| Real | 45% | 30% | 5% | 20% |
| Pseudo | 20% | 45% | 10% | 25% |
| **Evolution** | **No degradation** | **Minor deg.** | **Substantial deg.** | |
| Real → pseudo | 50% | 20% | 30% | |

and area shapes are conserved for half of test images, or lowly altered for 20% of them. For the 30% remaining images, the degradation is substantial and is a consequence of a too blurry generation. Such analysis is helpful to assess the clinical significance of the approach and highlight improper images being hidden when looking at quantitative metrics only.

## 4. Conclusion

In this work, we illustrate the potential of virtual histology generation from weakly paired MR. Our approach relies on a dual synthesis concept that composes two cycle-consistent generative adversarial networks. The only required input is a weakly paired MR imaging, on which unsupervised image-to-image translation is performed. Therefore, the clinical scope is very broad, giving the oncologist a realistic biological assessment of tumor environment prior to treatment.

This study, performing well in the prostate, represents the founding stone of the very ambitious virtual histology project and still needs additional features and enhanced performance. To produce substantial clinical value, objectives and challenges are: improvement of resolution towards conventional WSI granularity to extract tumor heterogeneity and fully bridge the technological gap between MR and histology, generalization to other locations with new types of tissue, and addition of an automatic tumor segmentation pipeline. In this way, a radiologist's assessment can be compared, towards multiple implications for cancer care, like margin improvement in radiotherapy - dose painting - or radiomics for immunotherapy.

## Appendix

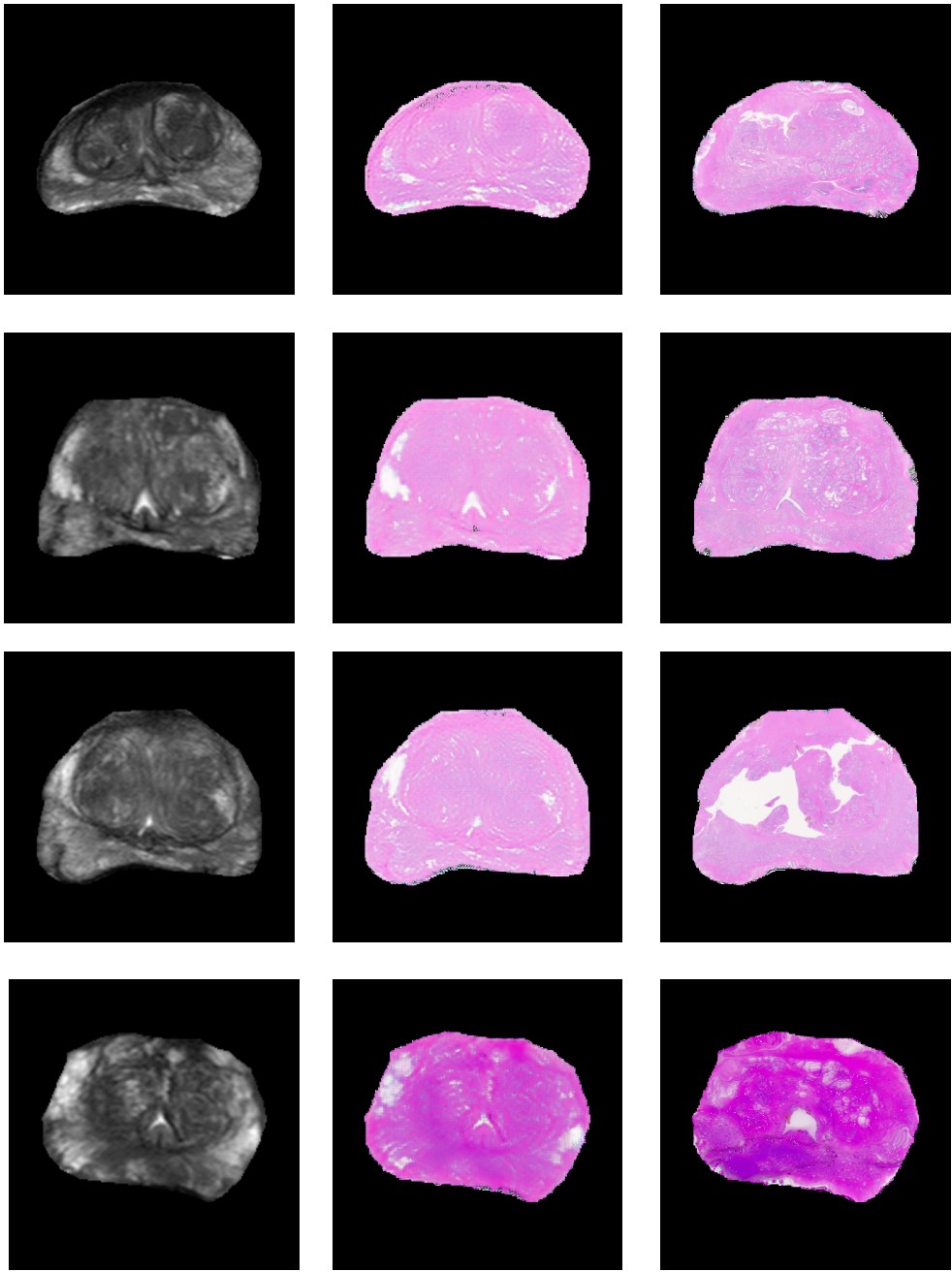

Figure 4: Samples from test set with (from left to right) : Original MR, Synthetic Histopathology, Original registered Histopathology. Generated examples adapt to stain change and reconstruct tissue even when ground truth histology has tissue collapse, giving additional information about cellular environment.

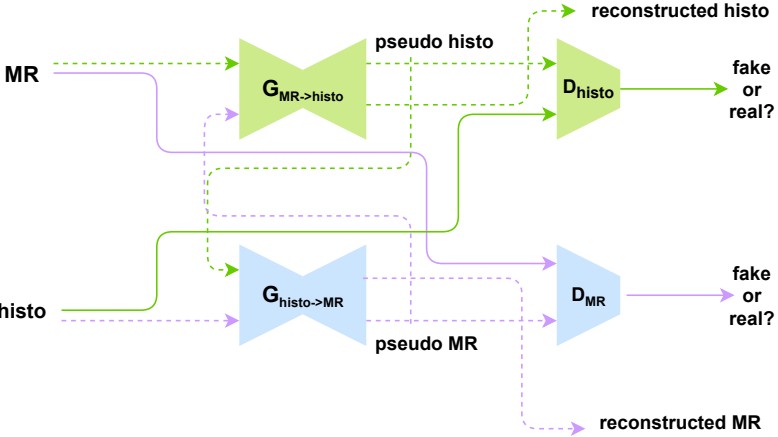

Figure 5: cycleGAN workflow with reconstruction consistency, two generators (similar to U-Net with residual blocks), two discriminators. This model outperforms conventional methods (encoder-decoder, conditional GAN) for unpaired image-to-image translation

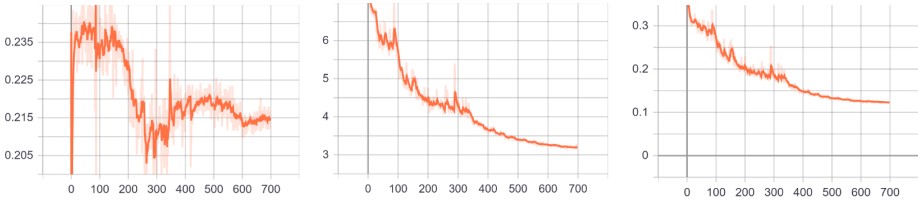

Figure 6: Evolution of training losses with epochs for (left to right) : Discriminator $D_H$, Generator $G_H$ and cycle-consistency loss. The training stops at 700 epochs, corresponding to the stabilization of models (early stopping). As the loss for discriminator is L2, we check that the final values are around 0.25, which is "random" performance, advocating for the good behaviour of the generator.

## Metrics definition

For two images x, y of size $n \times n$, we have:

$$MAE(x,y) = \frac{1}{n^2} \sum_{i=1}^{n} \sum_{j=1}^{n} |x[i,j] - y[i,j]| \tag{7}$$

$$PSNR(x,y) = 10 \log_{10}(\frac{max^2}{MSE}) \tag{8}$$

with $max$ the maximum value for a pixel and $MSE$ is the Mean Squared Error, meaning we sum the square of pixel differences compared to $MAE$.

$$SSIM(x,y) = \frac{(2\mu_x\mu_y + \epsilon_1)(2\sigma_x\sigma_y + \epsilon_2)(cov_{xy} + \epsilon_3)}{(\mu_x^2 + \mu_y^2 + \epsilon_1)(\sigma_x^2 + \sigma_y^2 + \epsilon_2)(\sigma_x\sigma_y + \epsilon_3)} \tag{9}$$

with $\mu$ the mean for an image, $\sigma^2$ the variance, $cov_{xy}$ the covariance between x and y, and $\epsilon$ an additional variable to avoid undefined quotient. Finally, for two datasets X, Y, with corresponding latent vectors (after Inception v3) following multidimensional Gaussian distributions $N(\mu_X, \Sigma_X)$ and $N(\mu_Y, \Sigma_Y)$

$$FID(X,Y) = |\mu_X - \mu_Y|^2 + tr(\Sigma_X + \Sigma_Y - 2\sqrt{\Sigma_X \Sigma_Y}) \tag{10}$$

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
