# OpenReview forum: "Magnetic Resonance Imaging Virtual Histopathology from Weakly Paired Data"
_MICCAI.org/2021/Workshop/COMPAY — COMPAY 2021_

### Official Review · Reviewer_VZWc · 2021-08-14
**Magnetic Resonance Imaging Virtual Histopathology**

**Rating:** 5
**Confidence:** 3

**Review:**

This paper presents an interesting idea about virtual histopathology from magnetic resonance imaging from weakly paired data, which is important in the clinical setting. The adopted method is straightforward based on cycle-consistent generative adversarial networks.
However, I have mixed feeling about this manuscript. The main concern comes from the valid evaluation and the demonstration of in a rigid accuracy. As mentioned by the authors, the tissue collapse and deformation pose hard challenges for valid evaluation. As we can see from Figure 3, there are significant differences between real images and synthetic ones. More importantly, I wonder if it’s feasible for solid evaluation considering the technical challenges during the clinical workflow. I would suggest authors can share more insights how this work can inspire more interesting studies in this direction.
In addition, for the quantitative comparison of results, no baseline comparison is provided. Although we can see some improvements from the ablation study modules, considering the deformation introduced during the tissue preparation, how we understand these evaluation results more reasonably?

---

### Official Review · Reviewer_5zS5 · 2021-08-24
**An exploratory study on generation of histology-like images from MRI imaging of the prostate**

**Rating:** 7
**Confidence:** 4

**Review:**

The authors present a study on generating images resembling low-resolution histology from MRI data.
For that task, they rely on publicly available data from TCIA to train two generative models
(i) an unsupervised generator using cycleGAN, and (ii) a self-supervised generator using registered data.
Overall, the paper is clearly written.
While I share the authors' enthusiasm about combining radiology and histological image data, my main critizism of the paper is that it the value added by the generated histology images is not clear. While the authors claim that the generated histology images are realistic, that realism only holds on a very coarse-grained level, namely for the gross regional composition depicted (i.e. the central, transition, peropheral and anterior zones). They do not show any information on tissue architecture or cellular growth patterns which are features central to any histological examination. In that light, it is not clear to me what added diagnostic information the generated 420x420 px pseudo histology images offer to the oncologist compared to the original MRIs. If they cannot show this value added, the authors should tune down their claims that the method as it stands is "giving the oncologist a realistic biological assessment of tumor environment prior to treatment", and that it reproduces "a more reliable representation of biological environment thanpost-resection ground-truth histology". This might be their vision but hardly seems justified by the results at this stage.

Minor points:
-A more detailed cohort description would be helpful (mean age, histologic subtypes included in the training cohort)
-The parameters used for augmentations  (rotations, in-plane rtranslation, stain contrasting) should be included explicitly

The manuscript might also benefit from showing a concrete example of image reistration as used for the self-supervised approach.

---

### Official Review · Reviewer_4Eh2 · 2021-08-25
**WSI synthesis from MRI images**

**Rating:** 5
**Confidence:** 3

**Review:**

The paper "Magnetic Resonance Imaging Virtual Histopathology From Weakly Paired Data" proposes an approach to generate WSIs from MR images.
The results of the proposed pipeline outperform the use of a single GAN or cycleGAN.The authors also involve a pathologist to evaluate the generated images against the original ones.

I think that the paper is at a too early stage for publication. The text is difficult to follow because of multiple flaws in the use of English. These affect the clarity of the paper.

I would encourage the authors to clarify the main hypotheses behind this work, clearly stating the objectives of the paper.
It is not clear from the introduction whether the final objective is generating WSI with cellular level definition. The authors claim the importance of WSI generation to obtain further details with non-invasive screening, but the paper does not comment on how this may be achieved in the future. The authors do acknowledge that more work is needed, but more comments about the future steps would be necessary to motivate this work.

The main contribution of this paper is the proposed pipeline, which improves the quality of the generation. I think the quantitative results show the benefits of this, but the user-tests are maybe a bit out-of-focus. Being presented first with the original image, the pathologist is biased towards recognizing the main structures also on the synthesized image.  I would suggest presenting the user with one image (either original and synthetize) and ask him to state whether the image is original or fake, and then annotate the structures. This approach would remove some bias. Besides, it seems to me that the structures annotated in Fig.3 are also visible from the MR image. I would also recommend improving the clarity of Table 2, what do the lines Real, Pseudo stand for? This is not clearly explained, nor in the text, nor in the caption.

Finally, more insights and discussion could have been provided in the paper.

---

### Decision · Program_Chairs · 2021-08-25

Accept